# Evaluation of Statistical Treatment of Left-Censored Contamination Data: Example Involving Deoxynivalenol Occurrence in Pasta and Pasta Substitute Products

**DOI:** 10.3390/toxins15090521

**Published:** 2023-08-24

**Authors:** Alessandro Feraldi, Barbara De Santis, Marco Finocchietti, Francesca Debegnach, Antonio Mandile, Marco Alfò

**Affiliations:** 1Department of Statistical Science, University of Rome “La Sapienza”, 00185 Rome, Italy; marco.finocchietti94@gmail.com (M.F.); mandile.1709269@studenti.uniroma1.it (A.M.); marco.alfo@uniroma1.it (M.A.); 2Department of Food Safety, Nutrition and Veterinary Public Health, Italian National Institute of Health, 00161 Rome, Italy; francesca.debegnach@iss.it

**Keywords:** left-censored data, limit of quantification, chemical contaminants, dietary exposure

## Abstract

The handling of data on food contamination frequently represents a challenge because these are often left-censored, being composed of both positive and non-detected values. The latter observations are not quantified and provide only the information that they are below a laboratory-specific threshold value. Besides deterministic approaches, which simplify the treatment through the substitution of non-detected values with fixed threshold or null values, a growing interest has been shown in the application of stochastic approaches to the treatment of unquantified values. In this study, a multiple imputation procedure was applied in order to analyze contamination data on deoxynivalenol, a mycotoxin that may be present in pasta and pasta substitute products. An application of the proposed technique to censored deoxynivalenol occurrence data is presented. The results were compared to those attained using deterministic techniques (substitution methods). In this context, the stochastic approach seemed to provide a more accurate, unbiased and realistic solution to the problem of left-censored occurrence data. The complete sample of values could then be used to estimate the exposure of the general population to deoxynivalenol based on consumption data.

## 1. Introduction

In order to assess the dietary exposure of population groups to a contaminant that can pose a risk for human health, information regarding contamination values for food commodities and food consumption data need to be collected. When performing chemical risk assessment for dietary exposure, food analyses provide positive contamination values (VAL, values in μg/kg) and non-detected, left-censored values that, for certain classes of contaminants, can be high in number [1,2]. With regard to the latter, the numerical value is not known, and the only information available is that it is below a given threshold, often representing a laboratory-specific feature. The parameters that describe non-detected values are particularly important for exposure estimations, as these correspond to the lower tail of occurrence data distribution. These parameters, which are indicative of the threshold value, are the limit of detection (LOD), defined as the lowest concentration level that can be determined to be different from zero [1,2,3], and the limit of quantification (LOQ), defined as the minimum concentration or mass of an analyte that can be quantified with a given confidence and a given analytical procedure [3,4,5]. Non-detected values are a real challenge for any statistical analysis involving occurrence data. Ignoring the undetected values and analyzing only the quantified ones will inevitably result in extensively biased and therefore unrepresentative estimates. In addition to the analytical thresholds of the LOD or LOQ, the percentages of non-detected values in a sample also depend on the type of contaminant and on the food category. Since a large portion of non-detected values may lead to computational difficulties in calculating percentile or mean values [6,7], decision tree recommendations are provided to guide the statistical treatment of datasets containing various proportions of non-quantified results [7].

Currently, different statistical methods are available to interpret non-detected values and model combinations of positive (VALs) and non-detected values. Traditionally, exposure assessment is performed through a deterministic substitution method, which is widely used in food risk assessment [8,9,10]. The substitution method is based on the replacement of non-detected values with a value of zero, the LOQ, or the LOQ/2 to obtain and represent the lower-bound (LB) and upper-bound (UP) or to approximate a medium-bound (MB) occurrence scenario, respectively [6,7]. This substitution approach has the advantage of being simple in its implementation. However, as any deterministic procedure for simple imputation, it also has some disadvantages. For instance, it may produce biased estimates of summary statistics (even when the maximum extent of such biases is assessed and does not exceed the difference between the upper and the lower bounds) [11]. Additionally, it does not consider the intrinsic variability of non-detected values that, through the process of imputation, are treated as observed, thus deflating the variability of any estimator based on such completed data.

Such drawbacks have been an object of study, and more recently the international interest has moved towards the application of stochastic approaches to impute (multiple) non-detected values in food contamination data. The literature includes several efforts to deal with the issue of non-detected values, especially via maximum likelihood (ML) estimation applied to parametric models with censored data [11,12]. For this purpose, several attempts have been made using marginal and pseudo-likelihood-based methods [13] by fitting truncated parametric (e.g., lognormal) distributions to observed data [14]; log-probit regression (LPR) [15]; or non-parametric methods, such as the non-parametric estimator of the survival function [12,16,17,18].

Quantification limits (and all the associated issues) are likely to be found in data obtained from analyses performed in different environmental contexts such as water, air, food, and even biological specimens like urine samples. For example, Jones (2018) applied complex linear regression models, which allowed outcomes and covariates to be linear combinations of left-censored data on polychlorinated biphenyls (a class of tocix industrial chemicals), for the estimation of means and covariance matrices [13]. A further study proposed a transformation to convert (multiple) left-censored data on water quality to right-censored data enabling the use of common survival analysis techniques [18]. Stochastic approaches impute non-detected values via a number of plausible values drawn from a given statistical model, and all approaches are implemented to understand the variability in the resulting estimates and overcome the uncertainty arising in the imputation of non-detected values. In this study, alternative approaches were employed to deal with non-detected values and with the presence of multiple LOQ values, aiming to analyze and handle contamination data related to deoxynivalenol (DON), a mycotoxin that is an agricultural contaminant occurring in cereals and all derived products (including pasta and pasta substitutes) produced by toxigenic fungi from the *Fusarium* genus. Mycotoxins are toxic substances produced by toxigenic micromycetes, such as those of the *Aspergillus*, *Penicillium*, and *Fusarium* genera, and when they occur in food or feed they represent a risk to human and animal health [19].

Contamination levels are highly influenced by weather conditions and agricultural practices [20,21], which, when unfavorable, may lead to severe *Fusarium* spp. infections affecting cereals, such as *Fusarium* head blight and European corn borer. Humid conditions at the time of flowering favor fungal proliferation on the spikelet and promote high levels of DON and other *Fusarium* toxins on crops. In its opinion published in 2017 [22], the European Food Safety Authority (EFSA) identified vomiting as the critical acute adverse effect of the sum of DON, its 3-glucoside, and the acetylated forms (3-Ac-DON and 88 15-Ac-DON) for pigs, farmed minks, dogs, cats, and human risk assessment.

Despite the limitations in the available human data, the EFSA established a group acute reference dose (ARfD) of 8 μg/kg body weight (bw) per eating occasion for the sum of DON, 3-Ac-DON, 15-Ac-DON, and DON-3-glucoside. Moreover, in the absence of data on chronic effects in humans, the EFSA recognized the reduced body weight gain in experimental animals as the critical chronic effect for human risk assessment and established a group tolerable daily intake (TDI) of 1 μg/kg bw per day for the sum of DON and its metabolites [22].

Conventionally, the analytical detection of DON is carried out by liquid chromatography (LC) coupled with UV detection (typically, wave length λ = 220 nm) [23,24]. In order to gain better selectivity, a purification step with an immunoaffinity column (IAC) is recommended to be used during the extraction of the mycotoxin. Nowadays, tandem mass spectrometry (LC-MS/MS) techniques are the most commonly used for the detection of mycotoxins in general, and DON in particular. The literature includes extensive examples of analytical methods applied to the analysis of DON, its metabolites, and even multiple mycotoxins in cereal-based foods [25,26,27,28,29].

In this study, DON contamination values detected in pasta and pasta substitutes were produced by a battery of laboratories that used several analytical techniques and/or different analytical methods, thus providing a range of LOQ/LOD values in the dataset. Therefore, the available data were not only censored, but the level and “quality” of censoring depended on the specific laboratory. To cope with this complexity, three different multiple imputation (MI) techniques for imputing non-detected values with multiple LOQ values were considered. Through the replacement of each non-detected value with multiple plausible contamination values, these techniques allowed us to obtain multiple sets of plausible contaminations. In order to achieve a completed contamination dataset, each set obtained by a given imputation procedure was merged with the VALs. The estimates obtained by imputation were compared to those derived from the application of standard, deterministic, and substitution methods on the same dataset, which could be used to define boundaries for the former.

As will be shown in the present paper, the proposed approach is appropriate for processing complex datasets with a high number of non-detected values and a range of different LOD/LOQ values.

## 2. Results

### 2.1. Exploratory Statistics

The number of LOQ values present in the analyzed dataset and the exploratory statistics for the data on DON contamination are reported in Table 1. For each LOQ stratum, the number of contaminated samples; the corresponding percentage, mean, median, and standard deviation for the measured values; the number and percentage of non-detected values; and the size and percentage (with respect to the global sample) of the corresponding subsample, are reported.

Considering VALs only (i.e., 131 values), the mean values in the total sample as well as in each subsample associated with each LOQ value were higher than the medians, denoting a positive skewness in the distribution of the contamination data related to DON in pasta and pasta substitute products. This result empirically supported the selection of lognormal, Weibull, and gamma as appropriate candidate distributions. To provide a clearer representation of the data at hand, Figure 1a,b show the empirical distribution of the contamination data. In particular, Figure 1a shows the CDF for the contamination data, with the crosses representing observed contamination data and each bin representing the cumulative frequency of non-detected values according to the value of the laboratory-specific LOQ. Figure 1b shows the distribution of VALs only, and the positive skewness in the distribution of data on DON contamination in pasta and pasta substitute products can be clearly observed. Prior to the imputation, the average contamination in the total sample was 207.11 μg/kg.

Before proceeding to exposure estimates, a choice had to be made as to the three candidates and the MA distributions. The last one was preferred because it summarized different behaviors that were specific to each candidate distribution without the need to refer to a more complex (in terms of the number of parameters) single parametric form. This procedure seems to produce a more honest measure of precision with a reduced bias when compared to a best fit model [30]. Additionally, inference based on a model averaging (MA) procedure seems to outperform that obtained using the best model strategy [31].

### 2.2. Comparing Stochastic and Deterministic Methods

In Table 2, a comparison of the results obtained by applying the (deterministic) substitution method with the stochastic multiple imputation approach for the distribution obtained by MA is shown in terms of some basic exploratory statistics. The application of both the proposed MI procedures and the standard deterministic substitution method in order to replace the 77 non-detected values led to a decrease in the estimate of the average contamination value with respect to that based on the VALs only (207.11 μg/kg). This was to be expected, as the non-detected values lay in the left tail of the contamination data distribution.

Moreover, Table 2 shows that the contamination distributions estimated through the three procedures were similar. More specifically, since the values to be imputed were below a certain threshold, as expected, the estimates of contamination mainly differed in the right tails of the distributions. A statistical comparison of the three distributions was performed using the Kullback–Leibler measure, a method that is widely used in order to quantify the dissimilarity between probability distributions [32]. The calculated Kullback–Leibler measures were consistently small, ranging from 0.03 to 0.96, which provided a confirmation of the similarity between the distributions obtained through the three procedures.

Figure 2 shows the density estimates for the completed contamination data provided by the stochastic MI procedure (All). As for the Gold-Standard and Single procedures (see Appendix A, Figure A1 and Figure A2, respectively), the positive skewness of the contamination data increased. This meant that DON contamination in pasta and pasta substitute products was effectively reduced when compared to the starting VAL distribution. With respect to the latter, after the MI and substitution methods, the distribution of contamination shifted to the left. This implied a reduction in the effective contamination scenario observed in pasta and pasta substitute products. In particular, a peak was shown between 0 and 50 μg/kg (see the top right portion of Figure 2), since the largest number of non-detected values (71%) belonged to the two largest strata, those associated with LOQ values equal to 26 and 50 μg/kg (see Table 1).

The estimated average contamination, the minimum and maximum values, and the quantiles provided by the suggested procedures were compared with the corresponding values obtained by the standard substitution method. This method is typically used because it is easy to understand and apply. It includes the LB, where the true contaminations are set equal to the minimum value that a non-detected value may assume (in this case zero), as well as the UB, where the true contaminations are all set equal to the corresponding LOQ, which represents their possible maximum value.

As shown in Table 2, the distributions of data on DON contamination in pasta and pasta substitute products derived from the stochastic multiple imputation approach (All, Gold-Standard, and Single) were included in LB and UB boundaries.

Figure 3 and Figure 4 show the estimated average DON contamination in pasta and pasta substitute products and the corresponding 95% confidence interval (CI) based on the B = 100 imputed datasets, according to the three procedures. Estimated using the All procedure, the average level of DON contamination in pasta and pasta substitute products was approximately 139.4 μg/kg, with a 95% confidence interval ranging from 137.0 μg/kg to 141.9 μg/kg. The Gold-Standard procedure yielded a slightly higher mean estimate of 145.3 μg/kg. The estimates from the Single procedure (mean of 137.7 μg/kg) closely aligned with those obtained from the All procedure. Finally, the lower- and upper-bound scenarios (of the substitution methods) were included in this 95% CI obtained by the proposed stochastic procedures.

## 3. Discussion

The strength of the substitution methods lies in the fact that they can be easily used to screen the levels and evaluate which of the components in the assessment path are likely critical. However, it has been widely recognized that the substitution method is biased. The bias is a function of the true data variability, the percentage of non-detected values, and the sample size. Moreover, the substitution method ignores the variability due to uncertainty in the real non-detected values that are imputed. In other words, whether a sample contains 1% or 90% detected values does not drive the way the non-detected values are treated. This applies even in cases where the two types of samples have different features [11], with the most critical situation being that with the highest number of non-detected values and the presence of multiple LOQ values [11,12].

Stochastic approaches introduce more realism, since they are based on fitting distributions to VALs and thus, as opposed to the deterministic approach, take the variability of the data into account when imputing non-detected values. Further, they may help to provide estimates of the variability ascribed to each imputation, whereas in the substitution method this is completely neglected. By using a stochastic approach, each missing value is imputed by B potential values taken from the selected distribution. Therefore, the B estimated values of contamination and incidence represent a set of scenarios that allow one to derive measures of synthesis (e.g., averages) and variability, such as 95% CIs.

### The Best Stochastic Procedure

The results of the exploratory statistics obtained by the procedure All, for which the imputation of non-detected values was based on the whole contamination sample, were the most reliable, as they took into account the whole left-censored contamination dataset. In the Gold-Standard procedure, the results depended on the stratum that was selected to be the best-fitted to the observed distribution. The selection could have limited the ability of the estimates to capture the real variability in the data on DON contamination in pasta and pasta substitute products, since it may not have reflect the variability and number of VALs in those strata that were characterized by a different LOQ value. Therefore, through the selection of the stratum with an LOQ equal to 50 μg/kg, the exploratory statistics for the contamination estimates provided by the Gold-Standard procedure proved to be slightly different from those obtained through the other two stochastic procedures (see Table 2). The Single and All procedures provided quite similar outputs. This was probably due to the fact that these two techniques had similarities, as they both considered all the contamination data to fit the candidate distributions. However, the Single procedure did not allow the imputation of non-detected values for the strata with no VALs (whose presence was mandatory in order to estimate the density distribution), and it did not make it possible to compute the likelihood for the stratum with an LOQ equal to 500 μg/kg. In addition, when the number of VALs was lower than 4 (as occurred in the stratum with an LOQ = 92.5 μg/kg), the estimated distribution was not informative and it was not sufficient for drawing imputations for non-detected values.

## 4. Conclusions

Three different procedures for the multiple imputation (MI) of left-censored data on DON contamination in pasta and pasta substitute products, which included a number of both detected (VALs) and non-detected values, were proposed. As a first output, the MI carried out through model averaging was to be preferred to the best fit model. This was due to its ability to produce estimates with a reduced bias as well as to summarize different behaviors that were specific to each candidate distribution. The procedure provided suitable imputations for pasta and pasta substitute products that were in agreement with each other and with those obtained by the deterministic estimates (LB and UB). In fact, the substitution method provided extreme scenarios for data on DON contamination in pasta and pasta substitute products, whereas MI provided contamination scenarios that were between the deterministic boundaries, which were too high (UB) or low (LB), respectively. Additionally, MI, which was based on fitting a specific parametric distribution, allowed us to account for (and estimate) the variability in the obtained contamination scenario due to imputation, which could not be achieved via deterministic methods. Among those proposed in this paper, the All procedure seemed to be the most appropriate.

In conclusion, MI provided suitable imputations and reliable contamination scenarios, especially for the All procedure performed via MA. However, the stochastic approach, which supplied distributions to be used for probabilistic calculations, should be considered complementary to the deterministic method rather than a replacement, since the latter led to conservative estimate calculations and offered an immediate assessment of the contamination boundaries.

Finally, although distributions and related mixtures other than lognormal, Weibull, gamma, and MA could be considered, the proposed approach was found to be appropriate for processing complex datasets of occurrence for contaminants with a high number of non-detected values, where the additional issue of a range of LOD/LOQ values makes the data handling even more challenging.

## 5. Materials and Methods

### 5.1. Occurrence Data

The available occurrence data for DON in pasta and pasta substitutes were gathered from the outcomes of the Italian National Official Control Plan for mycotoxins, which is the annual control plan (CP) requested by the European Commission from Member States. The main purpose of this is to verify the application of the rules and the functioning of national control systems. The CP is set each year by the Italian Ministry of Health to monitor and survey certain contaminants in food products throughout the national territory. The food sampling plans are defined with the support and collaboration of regions and provincial authorities, the “Direzione Generale per l’Igiene e la Sicurezza degli Alimenti e la Nutrizione” (DGISAN, Italian Ministry of Health, Rome, Italy) and the Italian National Reference Laboratory for mycotoxins in food and feed (NRL, Italian National Institute of Health, Istituto Superiore di Sanità), and they are approved by the “Coordinamento Tecnico Interregionale” (Technical Committee for the Heath Commission of the State-Region Conference). The annual CP provides guidance to regional and autonomous provincial authorities on the activities to be carried out by the official controlling body for the detection of mycotoxin occurrence (including DON) in food. It outlines and coordinates the actions for the verification of compliance with EU legislation and promotes the assessment of consumer exposure to contaminants.

The available dataset included data gathered and registered in the years 2016, 2017, and 2018. The pasta and pasta substitute samples were drawn at the final stage of the production and distribution of foodstuffs. The concentration levels of the contaminant in food were reported in terms of μg of DON per kg of food (μg/kg).

The samples were analyzed and the data were recorded by several official laboratories, distributed over the whole Italian territory. All laboratories operated under a quality control system following the ISO EN 17025 standard; they were equipped with different instruments for the detection and quantification of DON and used different analytical methods, for which different LOD/LOQ values were applied. The sample consisted of 208 contamination data for DON in pasta and pasta substitute products. Among these, 77 values (37.0%) were non-detected. The data collected came from laboratories distributed over eleven Italian regions and included six different LOQ values. Table 3 reports the distribution of DON occurrence, the production methods, the LOQ values, and the number of detected and non-detected contamination values reported by each laboratory.

### 5.2. The Proposed Stochastic Approach

In order to impute non-detected values, specific candidate parametric distributions were considered and fitted by maximum likelihood to the left-censored contamination data. After obtaining estimates for the parameters that indexed the parametric distributions, these were plugged in and used to draw plausible contamination values for the non-detected values. The whole (multiple) imputation procedure can be described by the following steps:Consider a (possibly wide) family of candidate distributions;Estimate parameters for each candidate distribution;Assess the quality of fit and select the best distribution in terms of fit;Model-average the candidate distributions with weights proportional to penalized likelihood criteria;Impute non-detected values (three techniques) by drawing several (i.e., 100) values from the model-averaged distribution for each non-detected value.

STEP 1. Specify the (possibly wide) family of candidate distributions

The contamination values of DON in pasta and pasta substitute products were considered to form an independent and identically distributed (iid) sample drawn from a known parametric distribution. The contamination values were assumed to be independent across food commodities, and therefore they could be modeled by univariate distributions. Contamination measurements usually show a skewed distribution. These distributions are particularly common when VAL data are at hand, as they are characterized by large variances, a high proportion of non-detected values, and multiple LOQ values. The lognormal distribution properly describes VALs, and it is parsimonious in the sense that it is indexed by a two-dimensional (mean, variance) parameter [33,34]. For these reasons, it was selected as the starting point while defining the set of candidate distributions [35,36,37,38]. In addition to the lognormal model, the Weibull and gamma distributions were also considered in the analysis, as they represent two further parametric models that are often used to model contaminant data in food [39,40,41,42]. 

STEP 2. Estimate parameters for each candidate distribution

For each candidate distribution, the problem of estimating the parameters can be summarised as follows: a sample of observations $y,y_{2},\ldots ,y_{n}$ of a quantitative variable (representing data on DON contamination) was given, and it was assumed to be drawn from a probability density function (PDF) f(x,θ), which could either describe a lognormal, Weibull, or gamma random variable, where θ is the vector of parameters that indexed such a density. Since the parametric form of the distribution was known, in order to use such a distribution, the vector of parameters needed to be estimated. In this context, estimation was carried out based on the maximum likelihood (ML). The likelihood function, which represents the objective function to be maximized, is defined as the joint probability density function evaluated at the sample points:(1)$$L\left(y_{1},y_{2},...,y_{n},\theta \right)=\prod_{i=1}^{n}f\left(y_{i};\theta \right),$$
while the maximum likelihood estimate (MLe) is defined as the value θ that maximizes the likelihood function.

In order to find this estimate, standard calculus methods (solving likelihood equations obtained by setting partial derivatives equal to 0 and checking that the matrix of second derivatives is negative-definite) are usually employed. However, when left-censored data are at hand, more complex methods are needed. ML estimation in the context of data left-censored by an LOQ [11,12] should consider the fraction of values above the LOQ and, therefore, the fraction of data below the LOQ, as well as the distribution for measured VALs. To summarise, let δ denote an indicator that is equal to 1 when the observation is noted and 0 when it is not. When δ=0, the information is that the value is below the threshold, i.e., *y* < LOQ. Further, let F(y;θ) denote the cumulative distribution function (CDF), associated with f(x,θ), for the selected parametric distribution with parameter vector θ. The MLe for θ is the value that maximizes the likelihood function *L* for such left-censored data:(2)$$L\left(\theta \right)=\prod_{i=1}^{n}{\left(f\left(y_{i};\theta \right)\right)}^{\delta_{i}}{\left(F\left(LOQ_{i};\theta \right)\right)}^{\left(1-\delta_{i}\right)}$$
where $LOQ_{i}$ denotes the LOQ for the i−th value. The censored observations could, in fact, contribute only through the CDF, as the only information they gather is that the corresponding values are not greater than the LOQ. Maximizing the likelihood function is equivalent to maximizing the corresponding log-likelihood function:(3)$$log\left(L\left(\theta \right)\right)=\sum_{i=1}^{n}\delta_{i}log+\left(1-\delta_{i}\right)log\left(F\left(LOQ_{i};\theta \right)\right)$$

Finally, the first and second derivatives of the log-likelihood with respect to θ are used to calculate the estimate for θ, usually via iterative Newton-type algorithms [43]. 

STEP 3. Assessing the quality of fit and selecting the best distribution in terms of fit

A number of model selection criteria can be used to choose the “best fitting” distribution, such as the Bayesian information criterion (BIC) [44] and the Akaike information criterion (AIC) [45]. While inspired by different motivations, they are both based on the idea that model fit can be measured by the maximized value of the log-likelihood function, once this has been penalized according to the model complexity. Therefore, the AIC is defined as:(4)AIC=−2logL(θy;δ)+2K
where logL(θy;δ) is the maximized value of the log-likelihood function for a candidate distribution, and *K* represents the corresponding number of parameters. The lower the AIC value, the better the fit. In this study, the AIC index was used as the model selection criterion. 

STEP 4. Model averaging

The MA step was performed using AIC-based weights [30]. MA starts from a set of plausible candidate statistical models (here, lognormal, Weibull, and gamma distributions were the considered distributions), and it combines such models in order to obtain a weighted average model where the weight associated with each candidate model is proportional to the corresponding measure of fit.

In detail, let us denote by *M* the family of candidate models, by $M_{j},j=1,\ldots ,K$ the individual model belonging to this family, and by $F_{j}$ the corresponding CDF. The procedure starts with the estimation of the natural parameters $\theta_{j}$ by ML; let us denote the corresponding covariance matrix $COV(\theta_{j})$ and the estimate of the p−th quantile $\xi_{p}$, p∈(0,1). In a direct approach to MA, the quantile $\xi_{p}$ is estimated for each candidate model, and the MA estimate is a weighted average of the K estimates. Based on model $M_{j}$, the MLe for $\xi_{p}$ is $\xi_{p,j}=F_{j}\left(p;\theta_{j}\right)$ [30]. The variance of the averaged estimate is approximated by the delta method:(5)$$VAR\left(\hat{\xi }\right)\approx \Delta F_{j}^{-1}{\left(p;\theta \right)}^{T}COV\left(\hat{\theta }\right)\Delta F_{j}^{-1}\left(p;\theta \right)$$
where the gradient $\Delta F_{j}^{-1}\left(p;\theta_{j}\right)$ can be estimated by $\Delta F_{j}^{-1}\left(p;\hat{\theta_{j}}\right)$. The MA estimate is defined as the weighted average ${\hat{\xi }}_{p,MA_{1}}=\sum_{i=1}^{K}w_{i}{\hat{\xi }}_{p,i}$, where the weights $w_{i}$ associated with each candidate model are defined by:(6)$$w_{i}=\frac{exp(-\frac{1}{2}\Delta_{i})}{\sum_{j=1}^{K}exp\left(-\frac{1}{2}\Delta_{j}\right)}$$
and
(7)$$\Delta_{j}=AIC_{j}-AIC_{min}$$

The model closest to the best fitting one is associated with the highest weights. 

STEP 5. Impute non-detected values (All, Gold-Standard, Single)

In the presence of a single LOQ, it is assumed that a specific q-th percentile corresponds to the LOQ value or each candidate model. For each non-detected value B (i.e., B = 100), random values $\left(u_{1},\ldots ,u_{B}\right)$ were drawn from a continuous uniform distribution defined on the segment 0 to *q*, representing the tail of the distributions associated with non-detected values. Each value $u_{1},\ldots ,u_{B}$ represents a potential quantile $q_{1},\ldots ,q_{B}\leq q$ for the distributions (lognormal, Weibull, gamma, and MA) associated with the contamination values. In order to perform imputation, such quantiles were back-transformed to values between 0 and the corresponding LOQ value. A number (B) of completed datasets were obtained, each of them corresponding to a set of draws for the non-detected values. However, as remarked above, contamination data were characterized by different LOQ values, as these were specific to the recording laboratory. In order to handle the problem of multiple LOQ values, three procedures for the MI of non-detected values were proposed (namely, All, Gold-Standard, and Single). The All and Gold-Standard procedures imputed non-detected values by taking all the different LOQ values collectively, whereas the Single procedure imputed non-detected contaminations stratified by each LOQ value taken individually (i.e., taking 6 strata, see Table 1). 

Procedure 1 (All).

The parameters for the candidate distributions were estimated using ML for left-censored data on the incomplete contamination sample, considered as a whole. This method did not take into account the laboratory from which the data originated. The basic idea of this procedure was to impute non-detected values according to the best fit to all the observed data and all considered distributions (including the MA distribution). The distributions were estimated on all contaminations, considered as a unique sample. 

Procedure 2 (Gold-Standard).

After stratifying the contamination data according to the LOQ values, the stratum with the largest number of data (VALs and non-detected values) was selected as the Gold-Standard stratum, as this stratum provided the most accurate major empirical evidence. Therefore, the parameters indexing the candidate distributions were estimated through ML on the contamination data belonging to the Gold-Standard stratum. This procedure based the imputation of all non-detected contamination on the distributions of the contamination estimated using data from the Gold-Standard stratum only. In pasta and pasta substitute products, the Gold-Standard stratum had an LOQ equal to 50 μg/kg, with n = 133 contaminations in total, (94 VALs, and 39 non-detected values). Thus, all candidate distributions (including MA) were estimated based on observed contamination data with an LOQ of 50 μg/kg. 

Procedure 3 (Single).

After stratifying the contamination data according to the LOQ values, the parameters indexing the candidate distributions were estimated via ML applied to contamination data from each stratum separately. The basic idea of this procedure was to impute the non-detected values, lower than the LOQ, considering the potential variability of the contamination for that stratum when compared with those of other strata. Then, imputed contamination data corresponding to each LOQ-specific stratum were joined as a unique completed contamination dataset.

Obviously, the imputation procedure was repeated in each case B = 100 times, and B completed datasets were obtained by the following procedure. Once parameters for candidate distributions were estimated according to one of the procedures previously discussed, the percentiles corresponding to each LOQ (i.e., 26 μg/kg, 50 μg/kg, 92.5 μg/kg, 100 μg/kg, 150 μg/kg, and 500 μg/kg) were computed for each candidate distribution. Thus, considering, for example, the stratum with LOQ = 26 μg/kg, which contained 16 non-detected values to be imputed, 100 values between 0 and the (distribution-specific) quantile corresponding to 26 μg/kg were randomly drawn for each non-detected value from the three candidate distributions. Afterwards, the corresponding quantiles as well as those obtained with a step of MA were computed. These quantiles were back-transformed (using the inverse sampling method) to contamination values in order to replace non-detected values below LOQ = 26 μg/kg. In this way, pseudo-random contamination values were drawn below the corresponding LOQ. This procedure was carried out for all the strata, each having a different number of non-detected values. Regardless of the imputation procedure selected (All, Gold-Standard, and Single), the set of imputed contaminations was then joined to the VALs. The multiple completed contamination datasets obtained by applying the All and Gold-Standard procedures consisted in 208 rows and 100 columns; each column referred to a completed dataset, where each of the 77 non-detected values was replaced by a set of reliable/plausible values drawn as discussed before from the candidate distributions, whereas the subset of 131 VALs was kept fixed. The procedure was repeated B = 100 times, and so the columns were filled in. As opposed to others, the Single procedure provided a smaller multiple complete contamination dataset (203 rows and 100 columns). Indeed, it was not possible to impute 5 non-detected values with an LOQ equal to 500 μg/kg, since the estimation of the candidate distributions is not possible for strata with no VALs. The phase of drawing several values lower than the corresponding LOQ for each non-detected value of DON in pasta and pasta substitute products is coherent with the Monte Carlo approaches used for sampling from a given theoretical (parametric) probability distributions [46].

### 5.3. Comparison between Deterministic Substitution Methods and Stochastic Approaches

To obtain contamination references as boundary levels to be compared with the results provided by MI, a substitution method (deterministic approach) was also applied. The contamination boundaries were defined as the LB and UB.

## Figures and Tables

**Figure 1 toxins-15-00521-f001:**
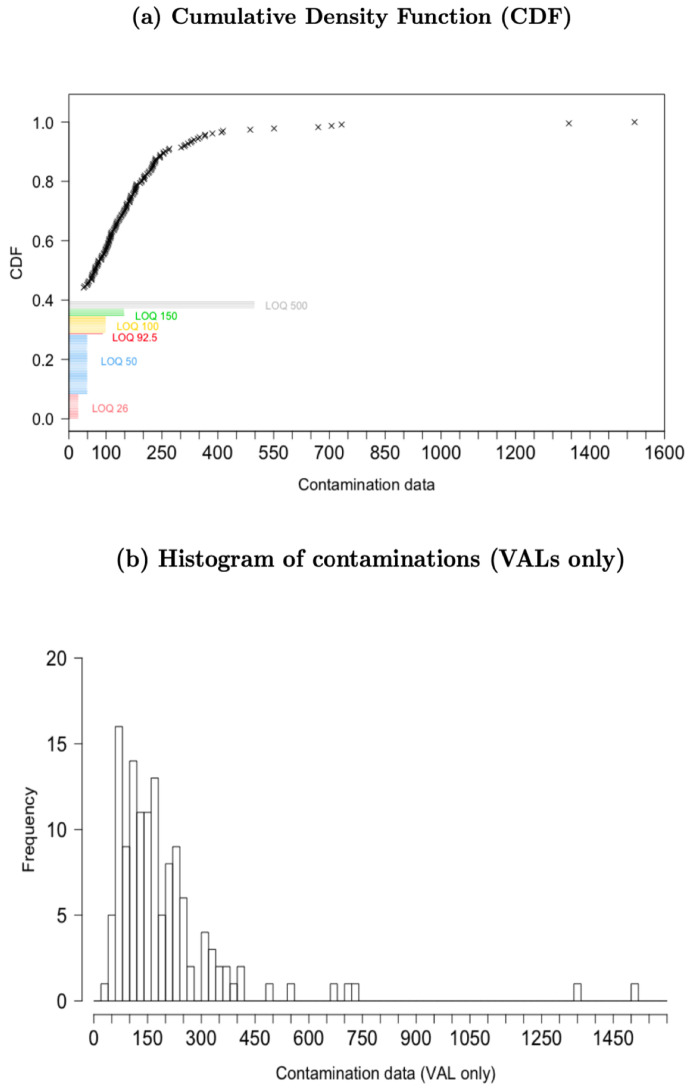
Distribution of observed data on DON contamination in pasta and pasta substitute products: (**a**) cumulative density function (CDF), (**b**) histogram of contaminations (VALs only).

**Figure 2 toxins-15-00521-f002:**
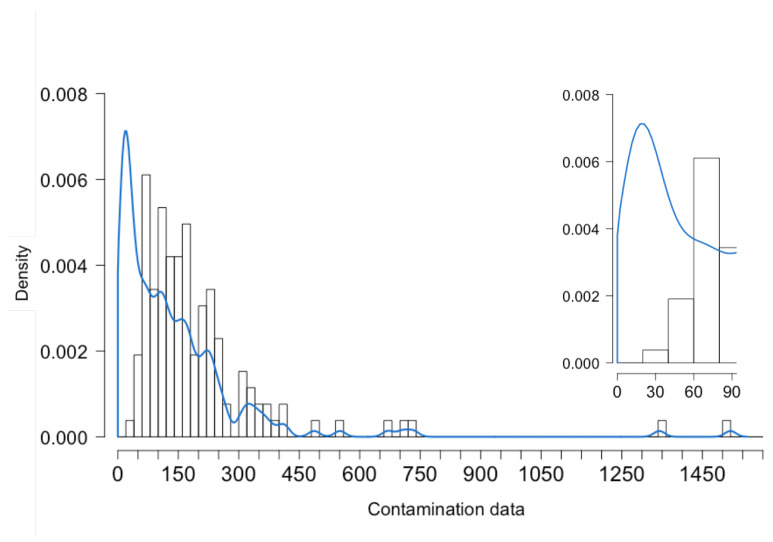
Density estimates of data on DON contamination in pasta and pasta substitute products applying All (MA only).

**Figure 3 toxins-15-00521-f003:**
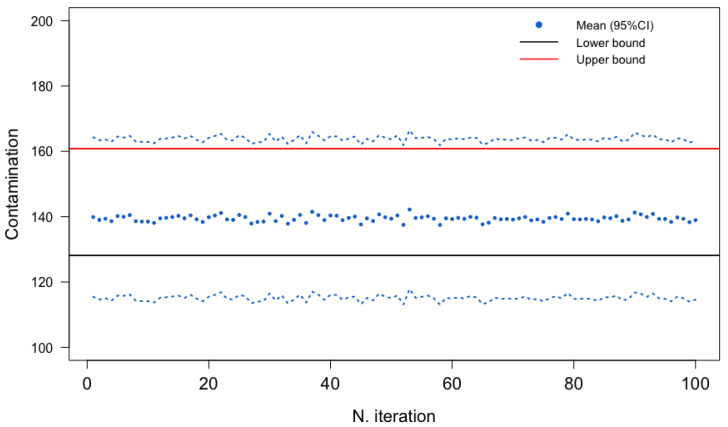
Mean estimate and 95% confidence interval (CI) for the mean of data on DON contamination in pasta and pasta substitute products across iterations (multiple imputation) (All procedure).

**Figure 4 toxins-15-00521-f004:**
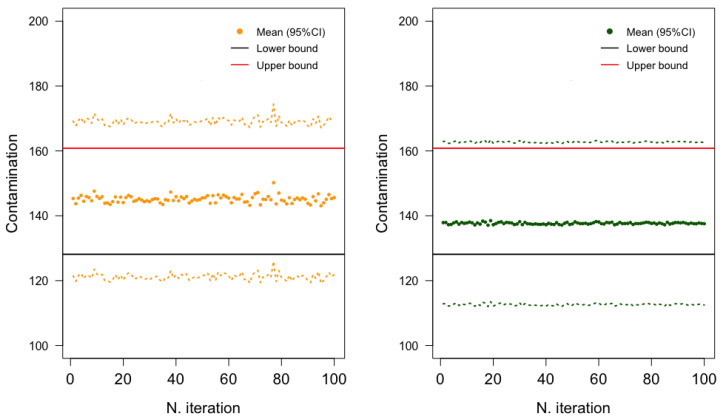
Mean estimate and 95% confidence interval (CI) for the mean of data on DON contamination in pasta and pasta substitute products across iterations (multiple imputation) (**left**: Gold-Standard procedure, **right**: Single procedure).

**Table 1 toxins-15-00521-t001:** Contamination values (μg/kg) of pasta and pasta substitute products by LOQ. Exploratory statistics and frequency distribution.

	Detected					Non-Detected		Total	
**LOQ (μg/kg)**	**n**	**%**	**Mean (μg/kg)**	**Median (μg/kg)**	**SD**	**n**	**%**	**n**	**%**
26	23	59.0	111.5	101.0	56.9	16	41.0	39	18.8
50	94	70.7	225.6	169.2	230.8	39	29.3	133	63.9
92.5	3	75.0	371.0	220.5	288.4	1	25.0	4	1.9
100	9	45.0	203.2	164.0	99.6	11	55.0	20	9.6
150	2	28.6	207.0	207.0	35.3	5	71.4	7	3.4
500	0	0.0	-	-	-	5	100.0	5	2.4

Abbreviations: n—number, %—percentage, SD—standard deviation.

**Table 2 toxins-15-00521-t002:** Exploratory statistics of the completed distribution for DON in pasta and pasta substitute products (μg/kg).

	CONTAMINATION VALUES					
**Procedure**	**Min**	**1st Quartile**	**Median**	**Mean**	**3rd Quartile**	**Max**
*All*	0.3	29.7	98.9	139.4	180.1	1519.6
*Gold-standard*	5.1	44.0	101.0	145.3	181.0	1519.6
*Single*	0.4	20.1	98.9	137.7	181.0	1519.6
*LB*	0.0	0.0	94.2	128.1	178.9	1519.6
*UB*	26.0	50.0	108.2	160.8	195.2	1519.6

**Table 3 toxins-15-00521-t003:** Distribution of DON occurrence, production methods, LOQ values, and detected/non-detected values across different laboratories.

Region	Production Method	LOQ	N. VAL	%VAL	N. LOQ
Basilicata	Unknown	50	1	100	0
		92.5	3	75	1
Emilia Romagna	Non-organic production	50	39	87	6
Friulia-Venezia Giulia	Non-organic production	100	8	73	3
Lazio	Unknown	26	23	59	16
Liguria	Unknown	50	2	67	1
Lombardia	Organic production	100	0	0	5
Piemonte	Unknown	50	2	50	2
		500	0	0	5
Puglia	Unknown	50	50	63	30
Sicilia	Unknown	100	0	0	3
Umbria	Unknown	150	2	29	5
Veneto	Unknown	100	1	100	0

Abbreviation: %—percentage.

## Data Availability

The data that support the findings of this study are available on request from the corresponding authors, Alessandro Feraldi and Barbara De Santis.

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
