# Peer review of "Evaluation of Statistical Treatment of Left-Censored Contamination Data: Example Involving Deoxynivalenol Occurrence in Pasta and Pasta Substitute Products"

_toxins, 2023, doi:10.3390/toxins15090521_

Round 1

Reviewer 1 Report

1. In Line 208, “STEP 5” should be on the following line, with “Impute non-detected values (All, Gold-Standard, Single)”.

2. Why not compare the Density estimates of data on DON contamination in pasta and pasta substitute products applying different procedures?  

3. Please compare mean estimate and 95% Confidence Interval (CI) for the mean of data on DON contamination in pasta and pasta substitute products by different procedures.

1. The language and details need to be perfected. For example, what is the meaning of “iid” in line 136?

Author Response

Reviewer 1. Comments and Suggestions for Authors

  1. In Line 208, “STEP 5” should be on the following line, with “Impute non-detected values (All, Gold-Standard, Single)”.

Necessary corrections done.

  1. Why not compare the Density estimates of data on DON contamination in pasta and pasta substitute products applying different procedures?

Thank you for highlighting this point. The density estimated according to the All procedure was reported in Figure 2. The density estimated according to the Gold-Standard and Single were reported in Figures A1 and A2 in the appendix. We have included the comparison of the density estimates of data on DON contamination in pasta and pasta substitute products for the different procedures in section 3.2, as follows:

Additionally, Table 2 shows that the contamination distributions estimated through the three procedures were similar. Specifically, since the values to be imputed were below a certain threshold, as expected, the estimates of contamination differed mainly in the right tails of the distributions. We performed a statistical comparison of the three distributions using the Kullback-Leibler measure, a widely used method to quantify the dissimilarity between probability distributions (Belov and Armstrong, 2011). The calculated Kullback-Leibler measures were consistently small, ranging from 0.03 to 0.96, which provided confirmation of the similarity between the distributions obtained through the three procedures.

  1. Please compare mean estimate and 95% Confidence Interval (CI) for the mean of data on DON contamination in pasta and pasta substitute products by different procedures.

In accordance with the Reviewer suggestion, Authors have reported the comparison of the mean and 95% confidence interval for the three procedures after Figure 3. Additionally, we have included Figure 4 which reports the mean estimate and 95% Confidence Interval (CI) for the mean of data on DON contamination in pasta and pasta substitute products across iterations (multiple imputation) for the Gold-Standard and Single procedures.

The following has been added:

Figures 3 and 4 show the estimated average DON contamination in pasta and pasta substitute products and the corresponding 95% Confidence Interval (CI) based on the B=100 imputed datasets, according to the three procedures. The average level of DON contamination in pasta and pasta substitute products, as estimated using the All procedure, is approximately 139.4 µg/kg, with a 95% confidence interval ranging from 137.0 µg/kg to 141.9 µg/kg. The Gold-Standard procedure yielded a slightly higher mean estimate of 145.3 µg/kg, while the estimates from the Single procedure (mean of 137.7 µg/kg) closely align with those obtained from the All procedure. Finally, the lower and the upper bound scenarios (of the substitution methods) are included in this 95% CI obtained by the proposed stochastic procedures.

Comments on the Quality of English Language

  1. The language and details need to be perfected. For example, what is the meaning of “iid” in line 136?

The manuscript has been checked for language editing. The meaning of iid is independent and identically distributed, as reported in the list of abbreviations. Following the suggestion of the reviewer we have included the meaning in the manuscript as follows:

iid (independent and identically distributed)

Reviewer 2 Report

This interesting paper describes an important work on handling the data of mycotoxin contamination to achieve more precise and valid results. Multiple Imputation have been extensively used for working with complex data in medicine, environment, agriculture etc. At the same time, there were no or few publications on using this approach for analyzing mycotoxin contamination in food and feed. Therefore, the paper presents an interest for readers and is scientifically sound, but there are several drawbacks should be improved before the paper will be accepted for publication.

1. I believe the Introduction should contain a more detailed description of the analyzed mycotoxin (DON), its producers and the risks they present for human and animal health. The methods used for the detection of DON in grain and food products are also could be mentioned. Also, the authors have to change Fusarium 'genera' to 'genus' (line 74).

2. I would like to know more about the samples used in the study (2.1.). The authors could include the information about the origin (from which laboratories obtained), methods of detection; DON content (for VALs); stage of pasta production. Probably, an ideal option will be to make a Supplementary table.

3. The abbreviations in Table 1 should be explained. I understood, what 'n', '%' or 'SD' mean, but it would be useful to decipher they below the table.

4. I think Results and Discussion should be divided into separate sections.

5. The reference list should be extended and include more literature devoted to use of Multiple Imputation for solving of different problems, e.g. agricultural (Chen et al., 2013); water industry (Umar and Grey, 2023); drinking water safety (Jones et al., 2014); toxin detection (Kitamura et al., 2023), etc. Also I have to note that the Reference list contain a lot of literature of 1960-80s, and could be improved by including more modern papers. 

English is easy to understand, but I believe the authors have to use a professional editing service to improve the text. I should note that there are several sentences could be re-written:

line 2: ...as being composed... instead of 'they are composed

line 74: 'Fusarium genus' instead of 'genera'

line  114: "All laboratories operate...'

line 307: 'In Table 2 we compare...'

Author Response

Reviewer 2. Comments and Suggestions for Authors

This interesting paper describes an important work on handling the data of mycotoxin contamination to achieve more precise and valid results. Multiple Imputation have been extensively used for working with complex data in medicine, environment, agriculture etc. At the same time, there were no or few publications on using this approach for analyzing mycotoxin contamination in food and feed. Therefore, the paper presents an interest for readers and is scientifically sound, but there are several drawbacks should be improved before the paper will be accepted for publication.

  1. I believe the Introduction should contain a more detailed description of the analyzed mycotoxin (DON), its producers and the risks they present for human and animal health. The methods used for the detection of DON in grain and food products are also could be mentioned. Also, the authors have to change Fusarium 'genera' to 'genus' (line 74).

Genera with genus replaced in line 74. More detailed description of the risk of DON for human and animal health as well as of the procedures to detect DON in grain and food products has been added as follows:

Concerning human and animal health, special attention goes to Mycotoxins, which are toxic substances found in various crops and derived foods, and produced by toxigenic micromycetes, such as Aspergillus, Fusarium, and Penicillium (Vconkova, 2003). Among these, Fusarium-toxins, with over 300 identified mycotoxins, are particularly hazardous to human health (Vconkova, 2003). In particular, a B-type trichothecene mycotoxin deoxynivalenol (DON) is commonly found in cereals such as wheat, barley, oats, rye, and maize. Contamination levels are highly influenced by weather conditions and agricultural practices (De Broeve et al., 2013; Sumarah & Blackwell, 2022), whose disfavor may lead to severe Fusarium spp. infections affecting cereals, such as Fusarium head blight and European corn borer. Humid conditions at the time of flowering favor the fungal proliferation on the spikelet (Joint FAO/WHO Expert Committee on Food Additives, 2011) and promote high levels of DON and other Fusarium-toxins on crops. In its opinion published in 2017 (EFSA, 2017), EFSA identified vomiting as the critical acute adverse effect for pigs, farmed mink, dogs and cats and for human risk assessment also for the sum of DON, its 3-glucoside and the acetylated forms (3-Ac-DON, 15-Ac-DON). Despite the limitations in the available human data, EFSA established a group acute reference dose (ARfD) of 8 µg/kg bw per eating occasion for the sum of DON, 3-Ac-DON, 15-Ac-DON and DON-3-glucoside. Moreover, in the absence of data on chronic effects in humans, EFSA recognized reduced body weight gain in experimental animals as the critical chronic effect for human risk assessment and established a group Tollerable daily intake (TDI) of 1 µg/kg bw per day for the sum of DON and its metabolites (EFSA, 2017). The analytical detection of DON was traditionally carried out by liquid chromatography (LC) coupled with UV detection (typically wave length λ= 220 nm) (Klötzel, 2005; MacDonald, 2005). In order to gain better selectivity, a purification step with immunoaffinity column (IAC) is suitable to be used during the extraction of the mycotoxin. Nowadays, tandem mass spectrometry (LC-MS/MS) techniques are the most used for the detection of mycotoxins in general and DON in particular. Literature shows extensive examples of analytical methods for the analysis DON, its metabolites and even for multi-mycotoxins in cereal food based (Sulyok, 2006; 2007; Varga, 2013; Malachovà, 2014; De Santis, 2017).

  1. I would like to know more about the samples used in the study (2.1.). The authors could include the information about the origin (from which laboratories obtained), methods of detection; DON content (for VALs); stage of pasta production. Probably, an ideal option will be to make a Supplementary table

Following the reviewer's suggestion, we have included a table (i.e. new Table 1), which reports the distribution of on DON occurrence, the production methods, the LOQ values and the number of detected and not detected values across laboratories. Details regarding the stage of pasta production were unavailable.

  1. The abbreviations in Table 1 should be explained. I understood, what 'n', '%' or 'SD' mean, but it would be useful to decipher they below the table.

The explanations for each abbreviation inserted in Table 1.

  1. I think Results and Discussion should be divided into separate sections.

Following the reviewer's indication, we have reorganized the Manuscript, providing two separated sections for Results and Discussion.

  1. The reference list should be extended and include more literature devoted to use of Multiple Imputation for solving of different problems, e.g. agricultural (Chen et al., 2013); water industry (Umar and Grey, 2023); drinking water safety (Jones et al., 2014); toxin detection (Kitamura et al., 2023), etc. Also I have to note that the Reference list contain a lot of literature of 1960-80s, and could be improved by including more modern papers.

Authors have extended the literature (adding more modern references) and included the references suggested by the reviewer as follows,

Imputation has been used also to address other problems, e.g. agricultural (Chen et al., 2013); water industry (Umar and Grey, 2023); drinking water safety (Jones et al., 2014); toxin detection (Kitamura et al., 2023), etc.

Comments on the Quality of English Language

English is easy to understand, but I believe the authors have to use a professional editing service to improve the text. I should note that there are several sentences could be re-written:

line 2: ...as being composed... instead of 'they are composed

line 74: 'Fusarium genus' instead of 'genera'

line  114: "All laboratories operate...'

line 307: 'In Table 2 we compare...'

The manuscript has been checked for language editing. Sentences has been rephrased as suggested.

Round 2

Reviewer 2 Report

The reviewed version of the MS contains more information about the methods used and the results obtained. Also, the reference list was improved and now contains all the corresponding references. 

At the same time, I would like to say that the authors have to modify the reference list according the Toxins' requirements (in terms of formatting)

Also, professional English editing is recommended.

I insist that the authors have to use a professional English editing service to make the MS text better in terms of grammar and style. 

Author Response

Authors thank the Editors and the reviewers for the careful work. Authors found all the comments relevant and they believe that review significantly contributed to improve the quality of the article.

Reviewer 2. Comments and Suggestions for Authors

I would like to say that the authors have to modify the reference list according the Toxins' requirements (in terms of formatting)

Also, professional English editing is recommended.

I insist that the authors have to use a professional English editing service to make the MS text better in terms of grammar and style. 

Authors are grateful to the reviewer for having highly recommended the linguistic review. Thorough English editing has been extensively done through all the text.

Reference list has been checked and amended for a number of references wrongly formatted.